# Goedel-Prover: A Frontier Model for Open-Source Automated Theorem Proving

**Yong Lin**[*†]  **Shange Tang**[*†]  **Bohan Lyu**[‡]  **Jiayun Wu**[‡]  **Hongzhou Lin**[§]

**Kaiyu Yang**[¶]  **Jia Li**[∥]  **Mengzhou Xia**[†]  **Danqi Chen**[†]  **Sanjeev Arora**[†]  **Chi Jin**[†]

## Abstract

We introduce Goedel-Prover, an open-source language model that achieves state-of-the-art (as of April 5 2025) performance in automated formal proof generation for mathematical problems. A key challenge in this field is the scarcity of formalized mathematical statements and proofs, which we address through the following approaches. First, we train LLMs to convert natural language math problems from the Numina dataset to equivalent formal statements in Lean 4. This process creates the dataset Goedel-Pset-v1, which includes 1.64 million formal statements. Next, we develop a large dataset of formal proofs by training a series of provers. Each new prover can prove many statements that previous ones could not, and these new proofs are added to the training set for the next prover. Finally, we obtain the dataset Goedel-Pset-v1-solved, which contains proofs for over 800K statements from Goedel-Pset-v1. Supervised fine-tuning (SFT) of DeepSeek-Prover-V1.5-Base on Goedel-Pset-v1-solved (i.e., no RL) yields a model that achieves a success rate of 57.6% (Pass@32) on miniF2F benchmark, surpassing the previous leader DeepSeek-Prover-V1.5 (trained using SFT + RL on a proprietary dataset) by 7.6%. On PutnamBench, Goedel-Prover-SFT successfully solves 7 problems (Pass@512), ranking first on the leaderboard. Further RL training (including DPO) improves Goedel-Prover-SFT's success rate to over 60% (Pass@32) on miniF2F.

To aid future research, we provide an extensive discussion of our training methodology and design choices. We also fully open-source our codes, models, dataset, and formal proofs for 29.7K problems in Lean Workbook, nearly doubling the 15.7K solved by prior provers.

## 1 Introduction

Recent advancements in large language models (LLMs) have demonstrated remarkable capabilities in reasoning tasks, especially in solving mathematical problems (Guo et al., 2025; Yang et al., 2024a). These models excel at reasoning through natural language, which we refer to *informal reasoning*. However, natural language-based reasoning is difficult to automatically verify by machines, which undermines the reliability of informal reasoning in practical applications. This also makes it more difficult to further improve the reasoning capabilities of language models. In contrast to informal reasoning, *formal reasoning* allows reasoning in a machine-verifiable format, opening up new possibilities for verification and automation. In particular, proof assistants such as Lean (De Moura et al., 2015; Moura & Ullrich, 2021), Isabelle (Paulson, 1994), and Coq (Barras et al., 1997) provide formal

---

[*]YL and ST contribute equally to this work. {yong.lin,shangetang}@princeton.edu

[†]Princeton Language and Intelligence, Princeton University.

[‡]Tsinghua University.

[§]Amazon. This work is independent of and outside of the work at Amazon.

[¶]Meta FAIR. KY served an advisory role. All experiments were conducted on PLI servers.

[∥]Numina.

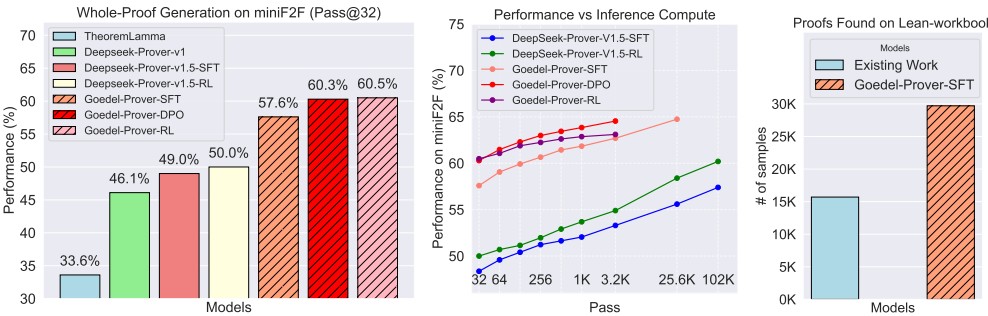

Figure 1: (**Left**) Pass@32 performance on miniF2F for whole-proof generation, compared to previous SOTA models. (**Middle**) A comparison of Goedel-Proverand DeepSeek-Prover-V1.5 on miniF2F performance across varying inference budgets, ranging from Pass@32, 64, 128, ..., to 4 × 6400. (**Right**) Goedel-Prover-SFT solves 29.7K problems in the Lean Workbook. In comparison, InternLM2.5-Step-Prover (Wu et al., 2024) and InternLM-Math-Plus (Ying et al., 2024b) collectively solved 15.7K samples.

languages that can express reasoning in a way that can be mechanically verified. Thus, it is of great interest to train LLMs to write proofs in these formal languages.

A significant challenge in training LLMs for theorem proving in formal languages is the scarcity of formalized math statements and proofs. Writing proofs for theorems expressed in formal languages is highly demanding and necessitates considerable domain expertise. Therefore, existing publicly available datasets for formal languages are limited in size. For example, the Lean Workbook (including Lean Workbook Plus) dataset (Ying et al., 2024a; Wu et al., 2024) comprises a total of 140K formal statements, where formal statements refer to problem statements in Lean without proofs. However, only 15.7K of these statements come with formal proofs, which were found by InternLM2.5-StepProver and InternLM-Math-Plus (Ying et al., 2024a; Wu et al., 2024; Ying et al., 2024b). Additionally, the Open Bootstrapped Theorems dataset (Wang et al., 2024) includes 107K statements with proofs sourced from Mathlib4 (mathlib4, 2023). However, Mathlib4 exhibits significant distribution shift from general problem-solving benchmarks, such as the widely used miniF2F (Zheng et al., 2021). See Section 4 for details.

In contrast to the scarcity of data in formal languages, there is a vast amount of math problems and solutions written in informal language. For example, Numina (Li et al., 2024a) includes 860K high-quality question and answer pairs sourced from MATH (Hendrycks et al., 2021), GSM8K (Cobbe et al., 2021), AMC (aop), AIME (MAA, 2024), the AoPS Forum (aop), Chinese K-12 Exams (Shao et al., 2024), World Olympiads, and synthetic data (Mitra et al., 2024). We start by training LLMs to formalize the problem statements in this dataset into Lean. To increase the diversity of the formalization styles, we train two formalizers. One is trained on informal and formal (I-F) statement pairs from Lean Workbook, while the other is trained on I-F statement pairs annotated by Claude-sonnet-3.5 (Anthropic, 2024). We use these two formalizers to formalize the statements and then employ LLMs to ensure that the formal statements preserve the content of the informal statements. Our efforts result in 1.64 million formal statements.

Using this extensive dataset of formal statements, we employ expert iteration (Polu et al., 2022) to train the prover to generate proofs. Notably, we train a model to generate complete proofs solely based on statements, without interacting with the Lean compiler during the generation process. This approach is referred to as the whole-proof generation method (Jiang et al., 2022; Wang et al., 2024; Xin et al., 2024a;b). At the beginning of the expert iteration, we generate 16 proof candidates using DeepSeek-Prover-V1.5-RL (the previous SOTA) for each formal statement in our dataset, and then we verify the correctness of each candidate using Lean compiler. The correct proofs are then collected to train our iter-1 prover based on DeepSeek-Prover-V1.5-Base. In subsequent rounds, we utilize our iter-$k$ prover to collect new proofs and add them to the training data. We then perform supervised fine-tuning

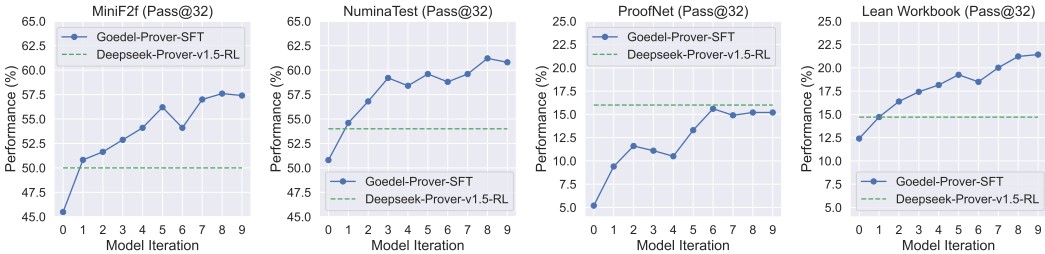

Figure 2: The figures show the performance of our model on the four datasets at each iteration. We gradually increase the size of the problem set and add more training data. The details of each iteration are shown in Table 8.

starting from DeepSeek-Prover-V1.5-Base for another round, resulting in the iter-$(k+1)$ prover. We conduct a total of 8 iterations and observe a consistent improvement starting from the first iteration.

We demonstrate that expert iteration, with large-scale formalized statements, can lead to SOTA performance in formal proof generation. Specifically,

- Our model, Goedel-Prover-SFT, outperforms DeepSeek-Prover-V1.5-RL (the previous SOTA model) by 7.6% on miniF2F, achieving a Pass@32 score of 57.6% (i.e., the prover generated 32 proofs for a problem, and 57.6% of the problems have at least one correct proof verified by Lean) compared to DeepSeek-Prover-V1.5-RL's 50.0%, as shown in Figure 1 (left). It consistently surpasses DeepSeek-Prover-V1.5-RL across all sampling budgets, including Pass@32, 64, and up to 25600, as shown in Figure 1 (middle).

- We have cumulatively solved 29.7K problems in Lean Workbook, significantly increasing the existing 15.7K proofs found by InternLM2.5-StepProver and InternLM-Math-Plus (Wu et al., 2024; Ying et al., 2024b), as shown in Figure 1 (right).

- Goedel-Prover-SFT solves 7 problems on PutnamBench by Pass@512[1], securing the **#1** position on the leaderboard (Table 2).

- We open source our codes[2], models[3 4 5 6], datasets[7 8], and the new proofs discovered[9] in the Lean Workbook to facilitate future research.

To understand the factors behind Goedel-Prover's strong performance, we provide an in-depth discussion of our training recipe, analyzing the effects of scaling up training data, the diversity introduced by autoformalization, correlations among datasets, and alternative data synthesis strategies. Furthermore, although our final model is trained purely through supervised fine-tuning, we also explore direct preference optimization (DPO) and reinforcement learning (RL) techniques built on top of it. Our Goedel-Prover-DPO and Goedel-Prover-RL achieve a success rate over 60% (Pass@32) on miniF2F. However, we also find that DPO and RL-trained models tend to overfit to "shortcuts" and benefit less from increased inference-time compute.

---

[1]We initially solved 8 problems on PutnamBench. However, after discussing with the authors of PutnamBench, we discovered that one of the problems was mis-formalized. Therefore, this problem is not included in our count, and we report a total of 7 problems here.

[2]https://github.com/Goedel-LM/Goedel-Prover

[3]https://huggingface.co/Goedel-LM/Goedel-Prover-SFT

[4]https://huggingface.co/Goedel-LM/Goedel-Prover-DPO

[5]https://huggingface.co/Goedel-LM/Goedel-Formalizer-32B-SonnetAnnotated

[6]https://huggingface.co/Goedel-LM/Goedel-Formalizer-32B-LeanWorkbookAnnotated

[7]https://huggingface.co/datasets/Goedel-LM/Goedel-Pset-v1

[8]https://huggingface.co/datasets/Goedel-LM/Goedel-Pset-v1-solved

[9]https://huggingface.co/datasets/Goedel-LM/Lean-workbook-proofs

## 2 Method

We begin by translating informal statements (expressed in natural language) into formal statements (represented in Lean). Using these formal statements, we iteratively train our prover with proofs generated by the prover and verified by the Lean compiler. The details of each step are elaborated in the following parts.

### 2.1 Statement Formalization

We first train the statement formalizers to translate informal statements in Numina into formal statements as shown in Figure 3. To enhance the diversity of formalized statements, we train two models to formalize informal statements.

- **Formalizer A:** We train the Formalizer A model using Formal and Informal (F-I) statement pairs sourced from Lean Workbook.

- **Formalizer B:** We employ Claude-sonnet-3.5 to formalize 230K statements from Numina. From this set, we extract 170K statements that successfully passed Lean compilation. These 170K F-I statement pairs are then used to train Formalizer B.

Both Formalizer A and B are trained using supervised fine-tuning with Qwen2.5-Coder-32B[10]. The training of these two formalizers takes less than 24 hours on 8 H100 GPUs. See Appendix A.1 for examples of formalized statements from these two formalizers, where we observe that even for the same problem, the style of the formalized statement can impact the prover's performance.

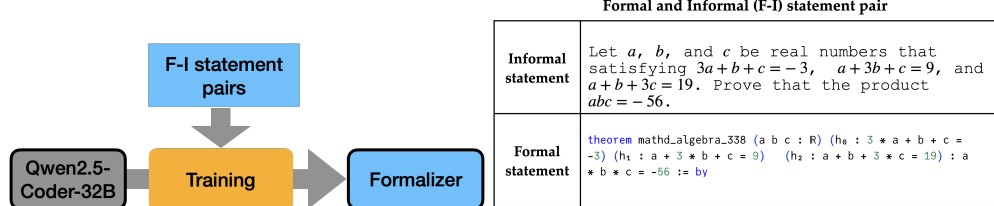

Figure 3: This figure illustrates the training of the formalizers. The term "F-I statement pairs" refers to pairs consisting of Formal and Informal (F-I) statements. An example is shown on the right-hand side. We train two formalizers, Formalizer A and B, using F-I statement pairs sourced from various origins.

**Quality assessment.** We employ two tests to assess the quality of the formalized statements. First, the formalized statement must conform to Lean syntax and can successfully compile, with the potential proof replaced by the placeholder ":= by sorry". This syntax check is known as the Compiling Correctness (CC) Test in the literature (Ying et al., 2024a). Second, the formalized statement must accurately capture the original informal problem, incorporating all assumptions, conditions, and implicit definitions. We refer to this second test as the Faithfulness and Completeness (FC) Test. For the FC test, we use Qwen2.5-72B-Instruct[11], details are presented in Appendix A.2. We also conducted a study comparing LLM and human ratings for the FC test and found an agreement rate of 85.7%.

In addition to formalizing the 860K open-sourced Numina (Li et al., 2024a) datasets, we also formalize a private 68K collection of math problems from Art of Problem Solving (AOPS), which has been collected and processed by the Numina group (Li et al., 2024a). Out of a total of 928K informal statements, 760K have two valid formalized statements generated by Formalizer A and B, while 123K contain only one valid formalized statement. After formalizing both the Numina and AOPS datasets, we further incorporate 140K statements

---

[10]https://huggingface.co/Qwen/Qwen2.5-Coder-32B
[11]https://huggingface.co/Qwen/Qwen2.5-72B-Instruct

from Lean Workbook, including Lean Workbook Plus. As a result, we have a total of 1.78M formal statements.

## 2.2 Expert Iteration

After obtaining a large collection of formalized statements in Section 2.1, we employ expert iteration to train the prover (Liu et al., 2024; Wu et al., 2024; Li et al., 2024b), which is illustrated in Figure 4. Specifically, we first utilize DeepSeek-Prover-V1.5-RL[12] to generate 16 proofs for each statement. We then verify these proofs with the Lean compiler. If at least one proof solves the statement, we retain one proof per statement. In cases where multiple proofs are available, we randomly sample one solution. These collected proofs are used for supervised fine-tuning (SFT) based on DeepSeek-Prover-V1.5-Base[13], resulting in the iter-1 prover. We continue this expert iteration process; each time, we use the iter-$k$ prover to generate answers and cumulatively collect correct solutions to train DeepSeek-Prover-V1.5-Base for the next iteration, the iter-$(k+1)$ prover. To prevent data leakage, we ensured that no statements in the training set were identical to those in the evaluation benchmarks. Refer to Appendix B for more details on each iteration.

We experiment with learning rates of $1 \times 10^{-4}$ and $5 \times 10^{-5}$, training for either 1 or 2 epochs. We use the packing trick (Tunstall et al., 2022) with a small batch size of 8 to speed up the training. In each iteration, the training time for 1 epoch is approximately 12 hours using 4 H100 GPUs. The inference time for the 1.78M statements set by Pass@16 is 6 hours, utilizing 64 H100 GPUs. Additionally, the verification time for these proofs requires 10 hours with 8,000 CPUs.

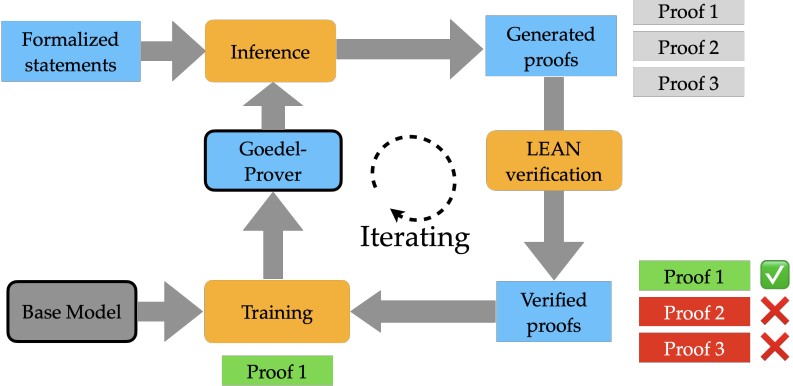

Figure 4: This figure illustrates the process of expert iteration. Each time, we utilize our iter-$k$ prover to collect new proofs and add them to the training data. We then conduct supervised fine-tuning starting from DeepSeek-Prover-V1.5-Base for another round, resulting in the iter-$(k+1)$ prover.

## 3 Results

**Benchmarks.** Following the works of (Wang et al., 2024; Xin et al., 2024a; Wu et al., 2024; Li et al., 2024b), we primarily use miniF2F (Zheng et al., 2021) as our main evaluation benchmark. We also track the problems solved by our prover in Lean Workbook (Ying et al., 2024a) and investigate the performance on ProofNet (Azerbayev et al., 2023) and PutnamBench (Tsoukalas et al., 2024). Additionally, we uniformly sample a subset from our formalized dataset to create a held-out evaluation dataset. Below, we provide descriptions of each dataset.

---

[12]https://huggingface.co/deepseek-ai/DeepSeek-Prover-V1.5-RL
[13]https://huggingface.co/deepseek-ai/DeepSeek-Prover-V1.5-Base

| Whole-Proof Generation Model | Pass | Performance |
|---|---|---|
| TheoremLamma (Wang et al., 2024) | 128 | 33.6% |
| Deepseek-Prover-V1 (Xin et al., 2024a) | 32 | 46.1% $\pm$ 0.5% |
| DeepSeek-Prover-V1.5-SFT (Xin et al., 2024b) | 32 | 48.2% $\pm$ 0.6% |
| DeepSeek-Prover-V1.5-RL (Xin et al., 2024b) | 32 | 50.0% $\pm$ 0.5% |
| Goedel-Prover-SFT | 32 | **57.6% $\pm$ 0.7%** |
| DeepSeek-Prover-V1.5-SFT (Xin et al., 2024b) | 3200 | 53.3% |
| DeepSeek-Prover-V1.5-RL (Xin et al., 2024b) | 3200 | 54.9% |
| Goedel-Prover-SFT | 3200 | **62.7%** |
| DeepSeek-Prover-V1.5-SFT (Xin et al., 2024b) | $4 \times 6400$ | 55.8% |
| DeepSeek-Prover-V1.5-RL (Xin et al., 2024b) | $4 \times 6400$ | 58.5% |
| Goedel-Prover-SFT | $4 \times 6400$ | **64.7%** |

Table 1: Whole-proof generation performance on miniF2F.

- miniF2F (Zheng et al., 2021) is a formal theorem proving benchmark, consisting of 488 problem statements (244 validation and 244 test problems) in Lean. The problems are drawn from high-school exercises, as well as high-school level competitions including the AIME, AMC, and the International Mathematical Olympiad (IMO). The original benchmark was released in Lean 3, and for our analysis, we use the version of miniF2F in Lean 4.9.0 provided by Xin et al. (2024a).

- ProofNet (Azerbayev et al., 2023) is a formal theorem proving benchmark of undergraduate-level mathematics, consisting of 371 problem statements in Lean (185 validation and 186 test problems). The problems are primarily drawn from undergraduate pure mathematics textbooks, covering topics such as real and complex analysis, linear algebra, abstract algebra, and topology. The original benchmark was released in Lean 3, and for our analysis, we use the version of ProofNet in Lean 4.9.0 provided by Xin et al. (2024a).

- Lean Workbook (Ying et al., 2024a) is a large-scale Lean 4 problem set formalized from natural language math problems (mainly from the forum AOPS), which consists of 140K statements in Lean 4. We also monitor the problems solved by our model during the expert iteration process. Notably, the problem set from Lean Workbook is included in this training, which is consistent with DeepSeek-Prover-V1.5 (Xin et al., 2024a) and InternLM2.5-StepProver (Wu et al., 2024).

- PutnamBench (Tsoukalas et al., 2024) is a formal theorem proving benchmark on competition mathematics problems sourced from the William Lowell Putnam Mathematical Competition years 1962 - 2023. PutnamBenchcomprises 644 Lean 4 statements, covering algebra, analysis, number theory, geometry, combinatorics, probability and set theory.

- NuminaTest. We randomly sample 250 statements from our formalized Numina dataset and use them as a held-out testing set. We refer to this subset as NuminaTest.

**Main results.** The performance on miniF2F is shown in Table 1. The Pass@32 performance of our Goedel-Prover-SFT is 57.6%, surpassing the previous SOTA open source model, DeepSeek-Prover-V1.5-RL, by 7.6%. We observe that our Goedel-Prover-SFT's Pass@32 is even better than DeepSeek-Prover-V1.5-RL's Pass@3200 by 2.7%. Furthermore, when both evaluated by Pass@3200, our model achieves 62.7%, surpassing DeepSeek-Prover-V1.5-RL's 54.9% by 7.8%. Figure 1 illustrates the inference time scaling curve for our Goedel-Prover-SFT, DeepSeek-Prover-V1.5-RL and DeepSeek-Prover-V1.5-SFT. Goedel-Prover-SFT demonstrates significant improvements over both DeepSeek-Prover-V1.5-RL and DeepSeek-Prover-V1.5-SFT across all inference compute budgets. Figure 2 illustrates the performance of our model during each iteration. Overall, we observe a relatively consistent improvement in performance across iterations.

**PutnamBench performance.** Goedel-Prover-SFT solves 7 out of 644 problems in Putnam-Bench (Pass@512), achieving the first place on the PutnamBench leaderboard. The previous SOTA method ABEL (Gloeckle et al.) solves 7 with a slightly higher inference budget

| Ranking | Model | Type | Num-solved | Compute (Pass) | Size |
|---|---|---|---|---|---|
| 1 | **Goedel-Prover-SFT** ⋄ | WPG | 7 | 512 | 7B |
| 1 | ABEL (Gloeckle et al.) | TS | 7 | 596 | 8B |
| 3 | **Goedel-Prover-SFT** ⋄ | WPG | 6 | 32 | 7B |
| 3 | InternLM2.5-StepProver (Wu et al., 2024) ⋄ | TS | 6 | 2×32×600 | 7B |
| 5 | InternLM 7B (Ying et al., 2024b) ⋄ | WPG | 4 | 4096 | 7B |
| 6 | GPT-4o | WPG | 1 | 10 | N/A (API) |
| 7 | COPRA (GPT-4o) (Thakur et al., 2023) | WPG | 1 | 1 | N/A (API) |
| 8 | ReProver w/ retrieval (Yang et al., 2024b) ⋄ | WPG | 0 | 1 | 0.6B |
| 9 | ReProver w/o retrieval (Yang et al., 2024b) ⋄ | WPG | 0 | 1 | 0.6B |

Table 2: Number of problems solved on PutnamBench statements (out of 644). Goedel-Prover-SFT achieves the first place in the leaderboard. The performance numbers for existing works are taken from the leaderboard. Here ⋄ inidicates open-source models. Type abbreviations: WPG = Whole-Proof Generation, TS = Tree Search Method.

| Formalization Model | miniF2F | ProofNet | NuminaTest | Average |
|---|---|---|---|---|
| Formalizer A only | 56.5% | 13.8% | 59.6% | 43.3% |
| Formalizer B only | 56.2% | **15.2%** | 60.0% | 43.8% |
| Formalizer A and B | **57.6%** | **15.2%** | **61.2%** | **44.7%** |

Table 3: An ablation study on using two formalizers to formalize the statements. Using statements formalized by both formalizers improves the model's performance, illustrating the value of diverse formalization styles.

(Pass@596) and InternLM2.5-Step-Prover (Wu et al., 2024) solves 6 (Pass@2 × 32 × 600). The performance is summarized in Table 2.

**Proofs found in Lean Workbook.** The Lean Workbook, which includes Lean Workbook-plus (Ying et al., 2024a; Wu et al., 2024), formalizes 140K high-quality problems sourced from AOPS and the Compfiles data. Currently, proofs for only 15.7K statements in Lean Workbook have been found and made open-source by InternLM2.5-StepProver (Wu et al., 2024) and InternLM-Math-Plus (Ying et al., 2024b). In contrast, our model has discovered a significantly larger set of proofs within Lean Workbook, cumulatively solving 29.7K problems, as shown in Figure 1 (right). We open-source all the proofs found by our model to benefit the research community.

**Tactic Usage and Proof Quality.** We conducted a detailed analysis of the proofs generated by Goedel-Prover-SFT to examine its tactic usage, proof quality, and the sources of its performance gains. We found that its most frequently used tactic is have, which is used to introduce and prove intermediate lemmas, indicating a capacity for structured, multi-step reasoning. This contrasts with baseline models that more heavily favor automation tactics like norm_num. Furthermore, Goedel-Prover-SFT employs a more diverse set of tactics. While powerful automation tactics like linarith and simp are utilized, their frequency of use is comparable to or even less than in baseline models. These findings suggest that the performance gains are due to more nuanced proof strategies and deeper reasoning capabilities, rather than simply exploiting built-in automation. See Appendix C for further details on the comparison of tactic usage and proof quality.

## 4 Dissecting the training recipe

**Scaling up the number of formal statements improves model performance.** Figure 5 shows the performance of provers (average on miniF2F, ProofNet and NuminaTest) trained on different sizes of the formal statement set. For each statement, the corresponding proof is obtained using Goedel-Prover-SFT. We observe a consistent improvement in model performance as the size of the statement set increases, underscoring the value of scale in training effective provers.

| Model | Training Dataset | miniF2F | ProofNet | NuminaTest | Average |
|---|---|---|---|---|---|
| Deepseek-RL | – | 50.0% | **16.0%** | 53.6% | 39.9% |
| Iter-6 prover | Iter-5 proofs | **56.6%** | 13.3% | **59.2%** | **43.0%** |
| Iter-6 prover | Iter-5 proofs + Mathlib4 | 54.1% | 15.6% | 58.8% | 42.8% |

Table 4: Incorporating Mathlib4 into the training data enhances performance on ProofNet but reduces performance on miniF2F and NuminaTest, suggesting distribution shift between Mathlib4/ProofNet and other datasets.

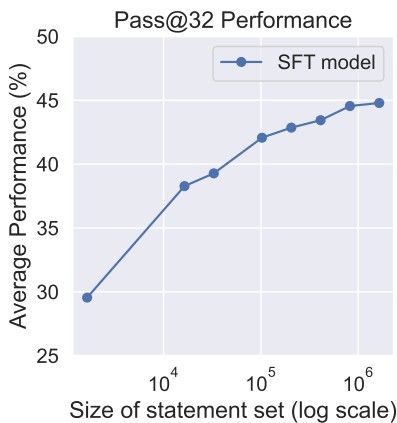

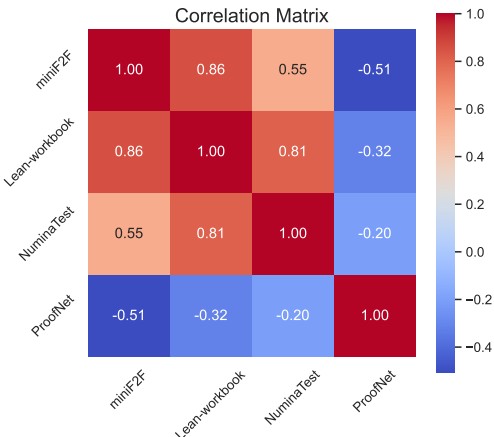

Figure 5: Model performance under different sizes of training statement set, illustrating the value of scale.

Figure 6: Correlation of model performance across datasets. ProofNet shows low correlation with the others.

**Increasing the diversity of formalization styles is beneficial.** Table 3 presents the performance of iter-8 provers trained on different formalization styles of statements, with proofs generated by the iter-7 prover. We find that a prover trained on a mixture of styles—combining statements produced by both Formalizer A and Formalizer B—outperforms provers trained on a single formalization style. This result suggests that exposure to diverse formalization patterns improves the model's generalization and reasoning ability.

We evaluate model performance across different training iterations and hyperparameter settings, and compute the correlation of performance across multiple datasets (see Figure 6). We observe that the model performance on ProofNet is negatively correlated with the performance on miniF2F, Lean Workbook, and NuminaTest. Furthermore, we investigate the effect of including Mathlib4 in the training data. As shown in Table 4, incorporating Mathlib4 improves performance on ProofNet but leads to a performance drop on miniF2F. These findings suggest a distribution shift between ProofNet/Mathlib4 and the other datasets. Specifically, Mathlib4 and ProofNet tend to focus on the manipulation of mathematical concepts, whereas datasets like miniF2F, Lean Workbook, and NuminaTest contain more Olympiad-style problems that emphasize complex reasoning over formal mathematical content. Illustrative examples are provided in Appendix D. Despite the observed distribution shift, we continue to include Mathlib4 in the training set from the sixth iteration onward, following the approach of DeepSeek-Prover-V1.5-RL (Xin et al., 2024b) and TheoremLamma (Wang et al., 2024), with the aim of enhancing the model's general capability across a broader range of mathematical domains. Additional training details can be found in Appendix B.

**Alternative approach for data synthesis.** In addition to autoformalizing statements and use the prover to provide proofs, we also explored alternative strategies for constructing training datasets, focusing on solving difficult problems by a divide-and-conquer strategy.

| Training method | Pass@32 (minif2f) | Pass@3200 (minif2f) | Average proof length | Average number of tactic "try" |
|---|---|---|---|---|
| **SFT** | 57.5% | 62.7% | 298 | 1.50 |
| **DPO** | 60.3% | 64.6% | 486 | 10.89 |
| **Length-penalized DPO** | 59.8% | 63.1% | 308 | 1.11 |
| **GRPO** | 60.5% | 63.1% | 355 | 5.16 |

Table 5: Models' behavior under different training methods. RL methods show improvement on miniF2F at Pass@32, but the improvement at Pass@3200 is limited . Furthermore, RL models are prone to excessively favor patterns such as `try`, which also causes the proof length to increase.

Inspired by Jiang et al. (2022), we implemented the following pipeline: (1) generate a proof for a formal statement using OpenAI's o1-preview model, (2) extract a high-level "sketch" of the proof and (3) apply DeepSeek-Prover-V1.5-RL to prove the subgoals provided by the sketch. If all the subgoals are successfully completed, we obtain a valid proof for the original problem. Implementation details are provided in Appendix E. However, this pipeline turned out to be ineffective in practice. When applied to the miniF2F validation set (244 problems), it successfully solved only 76 problems—considerably fewer than the 158 problems solvable by DeepSeek-Prover-V1.5-RL alone. Moreover, out of the 76 problems solved, only one is not solved by DeepSeek-Prover-V1.5-RL, implying that the marginal gain from this pipeline is limited.

**Exploring DPO and RL training.** We further explored DPO and RL training on top of Goedel-Prover-SFT. We implemented offline Direct Preference Optimization (DPO) (Rafailov et al., 2023) and online Group Relative Policy Optimization (GRPO) (Shao et al., 2024), implementation details are provided in Appendix F. Table 5 shows that although DPO and GRPO improve the model's Pass@32 performance, the average proof length grows substantially, and the frequency of certain patterns increases sharply. This phenomenon indicates that the model is overfitting to some syntactic patterns or "shortcuts", which is related to "reward hacking" (Chen et al., 2024). For example, the Lean tactic `try` allows trying a tactic and continuing execution regardless of whether it works or not. Although often harmless—and occasionally useful—its overuse can lead to ineffective proofs and substantial verification costs. The RL-trained model begins to excessively favor this pattern, ultimately impairing its reasoning and generalization capabilities.

Further experiments show that adding a length penalty during DPO training helps reduce this overfitting. However, we observe that scaling up inference-time compute yields significantly smaller gains for models fine-tuned with either GRPO or length-penalized DPO, compared to the SFT model. As shown in Table 5, these models achieve a 3% improvement over Goedel-Prover-SFT on Pass@32, but this gain diminishes when increasing inference-time compute—for example, at Pass@3200. This indicates that RL training may reduce output diversity, leading to less efficient inference-time scaling.

## 5 Related Work

**Automated theorem proving.** Automated theorem proving (ATP) is a long-standing problem in symbolic AI (Robinson & Voronkov, 2001). Traditional approaches represent theorems in first-order logic and prove them using decision procedures (De Moura & Bjørner, 2008; Barbosa et al., 2022) and search (Kovács & Voronkov, 2013; Schulz et al., 2019). The proof search has been enhanced by replacing handcrafted heuristics with machine learning techniques (Urban et al., 2011; Kaliszyk et al., 2018). However, approaches based on first-order logic struggle to scale to complex theorems and often do not yield human-readable proofs.

In recent years, learning-based theorem proving has undergone a significant transformation. A notable approach, introduced by Polu & Sutskever (2020), involves leveraging large language models to assist in theorem proving with proof assistants such as Lean (De Moura

et al., 2015; Moura & Ullrich, 2021) and Isabelle (Paulson, 1994). Follow-up research has explored various avenues, such as retrieving useful lemmas (Irving et al., 2016; Mikuła et al., 2024; Yang et al., 2024b), utilizing Monte Carlo tree search for proof discovery (Lample et al., 2022), and harnessing the capabilities of large language models (LLMs) for natural language reasoning (Jiang et al., 2022; Lin et al., 2024). Notably, Polu et al. (2023) was the first to apply expert iteration (Anthony et al., 2017) to theorem proving. This method alternates between two phases: (1) attempting to prove unsolved theorems and (2) enhancing the prover by incorporating newly discovered proofs into its training data. Expert iteration has yielded significant improvements in several recent provers (Wu et al., 2024; Xin et al., 2024b), including our Goedel-Prover.

Most automated theorem provers operate in a stepwise manner, generating individual proof steps that are then assembled into complete proofs using proof search algorithms. Recently, researchers have shown that generating entire proofs is feasible (First et al., 2023; Xin et al., 2024a; Wang et al., 2024). This approach avoids the costly search process, resulting in lower latency and potentially offering a more efficient use of computational resources during testing. While Goedel-Prover also generates whole proofs, our data and methodology can, in principle, be adapted to develop stepwise provers as well.

**Autoformalization and synthetic data generation.** The shortage of high-quality formal mathematical data poses a significant bottleneck in training theorem-proving models. While techniques like reinforcement learning may reduce the reliance on human-written proofs (Google DeepMind, 2024), there remains a need for a substantial number of formal theorem statements. A promising approach is to synthesize formal statements through autoformalization, where large language models (LLMs) translate informal mathematical statements into formal ones (Wu et al., 2022; 2024; Xin et al., 2024a;b).

DeepSeek-Prover (Xin et al., 2024a) and InternLM2.5-StepProver (Wu et al., 2024) have successfully implemented this strategy to formalize a large volume of statements into Lean for expert iteration. We adopt a similar approach. The difference is: while Liu et al. (2024) focuses on formalizing their internal dataset, we concentrate on formalizing the Numina dataset (Li et al., 2024a) alongside a privately collected dataset. Additionally, we train two formalizers to enhance the diversity of formalization styles, which we demonstrate to be beneficial in Section 3.

## 6 Conclusion and Discussion

We introduced Goedel-Prover, a new state-of-the-art open-source model for automated theorem proving. Our main contribution is a method to overcome the prevalent issue of data scarcity. By autoformalizing natural language problems and applying expert iteration, we created a massive dataset of over 800,000 formal statements with verified proofs. Our model, trained via supervised fine-tuning (SFT) on this data, achieves a 57.6% Pass@32 success rate on miniF2F, surpassing the previous leader by 7.6%, and ranks first on the PutnamBench leaderboard. We are open-sourcing our models, code, datasets, and the 29.7K new proofs we found for the Lean Workbook to spur further research.

While we explored reinforcement learning based algorithms (DPO/RL), we found that these methods, despite initial gains, encouraged "reward hacking" (e.g., overfitting to shortcuts like the `try` tactic) and scaled poorly with increased inference compute. This underscores the robustness of our SFT approach on large-scale, high-quality data.

Future work can build on this strong foundation by incorporating online interaction with the proof assistant for dynamic proof generation, by integrating external symbolic tools like SymPy to handle complex calculations, and by incorporating O1 (Jaech et al., 2024)/R1-like (Guo et al., 2025) long chain-of-thought capabilities to enhance the model's multi-step reasoning abilities when tackling complex theorems. By making our entire pipeline public, we aim to provide a powerful new resource to accelerate progress in open-source automated theorem proving. Detailed discussions are provided in Appendix G.

## Acknowledgments

CJ acknowledges the support from NSF-OAC-2411299, NSF-IIS-2239297, and Princeton AI Lab Seed Fund. SA acknowledge the support from Schmidt, Darpa, ONR, and NSF.

We thank Haoyu Zhao, Hubert Strauss and Suozhi Huang for their helpful discussions.

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

# Appendix

## A  Statement Formalization Details

### A.1  Examples of formalized statements

Table 6 presents two examples in which both Formalizer A and Formalizer B yield reasonable formalizations. However, our final prover exhibits varying performance on these formalized statements, highlighting the influence of formalization style on model effectiveness.

|  | **Example 1** | **Example 2** |
|---|---|---|
| **Informal Statement** | The function $f(x) = 2^{|x|} + 3x^2 + ax + 1$ is an even function, then $a$ equals $a = 0$. | If $x$ and $\log_{10} x$ are real numbers and $\log_{10} x < 0$, show that $0 < x < 1$. |
| **Formalizer A Output** | ```theorem lwb_style_example1  (a : ℝ)  (f : ℝ → ℝ)  (h₀ : ∀ x, f x = 2^abs x + 3 * x^2 + a * x + 1)  (h₁ : ∀ x, f (-x) = f x) :   a = 0   := by sorry```  Pass rate: 14/16 | ```theorem lwb_style_example2  (x a : ℝ)  (h₀ : Real.logb 10 x = a)  (h₁ : a < 0) :   0 < x ∧ x < 1   := by sorry```  Pass rate: 0/16 |
| **Formalizer B Output** | ```def IsEven (f : ℝ → ℝ) : Prop := ∀ x, f (-x) = f x  theorem sonnet_style_example1  (f : ℝ → ℝ)  (h_def : ∀ x, f x = 2^(|x|) + 3*x^2 + a*x + 1)  (h_even : IsEven f)  (a : ℝ) : a = 0 := by sorry```  Pass rate: 0/16 | ```theorem sonnet_style_example2  {x a : ℝ}  (h1 : x = 10^a)  (h2 : a < 0) :   0 < x ∧ x < 1 :=  by sorry```  Pass rate: 5/16 |

Table 6: Comparison of formalizer outputs for two examples. In Example 1, Formalizer A defines the "even function" directly by stating $f(-x) = f(x)$. In contrast, Formalizer B first introduces a function called "IsEven" and then defines the even function using "IsEven". Notably, our prover successfully solves the statements provided by Formalizer A but fails with those from Formalizer B. Example 2 is similar; however, our prover fails to solve the statement provided by Formalizer A but succeeds with the one from Formalizer B.

### A.2  Quality Assessment Details

For the FC test, we use Qwen2.5-72B-Instruct[14] with prompt shown in Figure 7. For each formalized statement, we generate four independent judgments, and the FC score is calculated as #{"Appropriate" in four Judgments}/4. For example, if the four judgments produced by Qwen2.5-72B-Instruct include three "Appropriate" and one "Inappropriate", the overall FC score is calculated as 0.75. We filter out formalized statements with an FC score less than 0.5.

For each informal statement in Numina, we generate eight formalized statements from each formalizer, resulting in 16 formalized statements per problem. Each statement undergoes the CC and FC Test, and we retain only those valid statements. We then randomly select one valid statement from each formalizer. For example, if five out of eight statements from Formalizer A and three from Formalizer B are valid, we randomly choose one from each. If a formalizer produces no valid statements, we exclude all its statements for that problem. The statistics for each test conducted on both formalizers are summarized in Table 7.

## B  Expert Iteration Details

The main training pipeline is illustrated in Section 2.2. When we implement the expert iteration algorithm, we gradually add the data. From iter-0 to iter-3, we gradually add

---

[14] https://huggingface.co/Qwen/Qwen2.5-72B-Instruct

```
You will receive a math problem consisting of its natural language statement and, in
some cases, a natural language proof or solution, along with its formal statement in
LEAN 4.

Please evaluate whether the formal LEAN statement appropriately translates the natural
language statement based on the following criteria:

1. Key Elements: The problem's essential components are correctly represented in LEAN
code.

2. Mathematical Accuracy: The translation preserves the accuracy of the mathematical
content.

3. Structural Fidelity: The translation aligns closely with the original problem,
maintaining its structure and purpose.

4. Comprehensiveness: All assumptions, conditions, and goals present in the natural
language statement are included in the LEAN translation.

Your answer should be in the following format:

Thought: [Your Answer]

Judgement: [Your Answer, one of {Appropriate, Inappropriate}]
```

Figure 7: Prompts for Faithfulness and Completeness (FC) Test.

| Model | Pass | Formalizer A | Formalizer B |
|-------|------|-------------|-------------|
| CC Test | Pass@1 | 76.74% | 88.48% |
| CC Test | Pass@8 | 95.93% | 98.59% |
| FC Test | Pass@1 | 48.06% | 80.42% |
| FC Test | Pass@8 | 88.01% | 97.22% |
| CC + FC Test | Pass@1 | 45.72% | 76.41% |
| CC + FC Test | Pass@8 | 82.33% | 95.78% |

Table 7: Quality assessment of the formalized statement

the statements formalized by Claude-sonnet-3.5. At iter-3, we train the Formalizer B and add the formalized statements generated by Formalizer B for iter-4 to iter-6. At iter-7, we begin to add the statements generated by Formalizer A. We also add Mathlib4 data into the training dataset for better ProofNet performance when starting from iter-6.

| Iteration | Statements | | Training Data | | |
|-----------|------------|-----------|---------------------|-------------------|----------|
| | Lean Workbook | Formalized | Lean Workbook Solved | Formalized Solved | Mathlib4 |
| Iter-0 | 140K | 0 | 20.6K | 0 | 0 |
| Iter-1 | 140K | 140K | 20.6K | 72.4K | 0 |
| Iter-2 | 140K | 270K | 23.0K | 128.7K | 0 |
| Iter-3 | 140K | 270K | 24.4K | 161.2K | 0 |
| Iter-4 | 140K | 882K | 25.4K | 425.8K | 0 |
| Iter-5 | 140K | 882K | 27.0K | 436.5K | 0 |
| Iter-6 | 140K | 882K | 27.8K | 443.2K | 104K |
| Iter-7 | 140K | 1.64M | 28.8K | 887.7K | 104K |
| Iter-8 | 140K | 1.64M | 29.7K | 915.7K | 104K |
| Iter-9 | 140K | 1.64M | 30.3K | 928.2K | 104K |

Table 8: Expert iteration details.

# C   Analysis of Tactic Usage and Proof Quality

In this section, we provide a detailed analysis of the proofs generated by Goedel-Prover-SFT, focusing on the diversity of tactics used, the reliance on high-level automation, and overall proof quality. This analysis studies whether performance improvements of our model might solely stem from better exploitation of Lean's built-in automation rather than from enhanced reasoning capabilities. We compare our model against the DeepSeek-Prover-V1.5-RL baseline on the miniF2F benchmark.

## C.1   Tactic Diversity and Frequency

To understand the strategic differences between the models, we first examine the most frequently used tactics. Table 9 lists the top 10 tactics for both Goedel-Prover-SFT and the baseline. Notably, the most used tactic for our model is have, which is crucial for constructing structured proofs by introducing and proving intermediate lemmas. This contrasts with the baseline, which most frequently uses norm_num, a powerful but more opaque automation tactic. This suggests that Goedel-Prover-SFT is not simply overusing automated tactics but is capable of producing more interpretable and structured proofs.

| Rank | DeepSeek-Prover-V1.5-RL | | | Goedel-Prover-SFT | | |
|---|---|---|---|---|---|---|
| | Tactic | Count | % | Tactic | Count | % |
| 1 | norm_num | 193 | 18.9% | have | 200 | 16.8% |
| 2 | have | 121 | 11.9% | linarith | 116 | 9.7% |
| 3 | linarith | 94 | 9.2% | norm_num | 96 | 8.1% |
| 4 | simp | 52 | 5.1% | simp_all | 54 | 4.5% |
| 5 | decide | 42 | 4.1% | simp | 49 | 4.1% |
| 6 | simp_all | 35 | 3.4% | only | 44 | 3.7% |
| 7 | omega | 30 | 2.9% | omega | 36 | 3.0% |
| 8 | only | 30 | 2.9% | nlinarith | 32 | 2.7% |
| 9 | with | 25 | 2.5% | ring_nf | 28 | 2.4% |
| 10 | nlinarith | 23 | 2.3% | Finset | 27 | 2.3% |

Table 9: Top 10 most-used tactics on miniF2F for Goedel-Prover-SFT and the baseline.

Furthermore, Table 10 provides a direct comparison of the usage of common high-level automation tactics. For most of these tactics, Goedel-Prover-SFT either uses them less frequently or at a similar rate as the baseline. This evidence further supports the conclusion that our model's improved performance comes from more nuanced and varied proof strategies rather than a simple over-reliance on automation.

| Tactic | DeepSeek-Prover-V1.5-RL | | | Goedel-Prover-SFT | | |
|---|---|---|---|---|---|---|
| | Count | % of Proofs | Avg per Proof | Count | % of Proofs | Avg per Proof |
| aesop | 0 | 0% | 0 | 1 | 0.7% | 0.01 |
| decide | 42 | 8.1% | 0.34 | 21 | 4.9% | 0.15 |
| linarith | 94 | 40.7% | 0.76 | 116 | 33.1% | 0.82 |
| nlinarith | 23 | 13.0% | 0.19 | 32 | 19.0% | 0.23 |
| norm_num | 193 | 46.3% | 1.57 | 96 | 43.7% | 0.68 |
| omega | 30 | 17.1% | 0.24 | 36 | 20.4% | 0.25 |
| ring | 11 | 8.1% | 0.09 | 20 | 9.9% | 0.14 |
| simp | 52 | 30.1% | 0.42 | 49 | 26.1% | 0.35 |

Table 10: Frequency of common high-level automation tactics used in successful proofs on miniF2F.

## C.2 Problem Difficulty and Proof Complexity

To assess whether Goedel-Prover-SFT is solving more difficult problems, we analyze the length of the generated proofs and the number of solved problems from challenging competition sources like AIME and AMC. As shown in Table 11, Goedel-Prover-SFT not only solves more problems overall but also generates a higher proportion of multi-step proofs (>3 lines) and solves more problems from the AIME and AMC categories. This indicates that our model is indeed tackling more complex problems that require deeper reasoning.

| Model | Total Solved | 1 line | 2-3 lines | >3 lines | AMC/AIME Solved |
|---|---|---|---|---|---|
| DeepSeek-Prover-V1.5-RL | 123 | 6 | 31 | 86 | 15 |
| Goedel-Prover-SFT | 142 | 10 | 37 | 95 | 19 |

Table 11: Analysis of proof length and problem difficulty on miniF2F.

Finally, we examined the overall diversity of tactics employed by each model, as summarized in Table 12. Goedel-Prover-SFT utilizes a significantly broader range of unique tactics, further demonstrating its ability to formulate more varied and sophisticated proof strategies. This wider tactical repertoire allows it to address a broader spectrum of mathematical challenges beyond those solvable by a more limited set of automated tactics.

| Model | Total Proofs | Total Tactics Used | Unique Tactics |
|---|---|---|---|
| DeepSeek-Prover-V1.5-RL | 123 | 1019 | 112 |
| Goedel-Prover-SFT | 142 | 1190 | 158 |

Table 12: Tactic diversity in successful proofs on miniF2F.

# D  More examples on style difference

## D.1  Mathlib4 and miniF2F

We observe a notable difference in the distribution of Mathlib4 compared to that of general problem-solving benchmarks, such as the widely used miniF2F (Zheng et al., 2021). For instance, miniF2F largely consist of competition and Olympic-style problems, which require complex reasoning, while only depending on a relatively small set of elementary facts about integers, real numbers, counting, and geometry. On the contrary the statements in Mathlib4 focus on the simple manipulation of advanced mathematical concepts. Figure 8 and 9 show the statement and proof in Mathlib4 and miniF2F respectively. It can be easily seen that both the statement and proof rely on pre-defined objects. Unlike miniF2F statements, the example in Figure 8 can not even pass the lean compilation, given that pre-defined objects are missing.

```
theorem ndrec_eq_ndrecC : @Acc.ndrec = @Acc.ndrecC := by
  funext α r motive intro a t rw [Acc.ndrec, rec_eq_recC, Acc.ndrecC]
```

Figure 8: A Mathlib4 example which relies on pre-defined objects @Acc.ndrec and @Acc.ndrecC

## D.2  ProofNet and miniF2F

The problems in ProofNet are primarily drawn from undergraduate pure mathematics textbooks, covering topics such as real and complex analysis, linear algebra, abstract algebra, and topology. These topics largely rely on the abstract and general formulations of mathematical definitions in Mathlib4 (mathlib4, 2023). We show two examples in Table 13 to illustrate the style difference between ProofNet and miniF2F.

```
/-- If $3a + b + c = -3, a+3b+c = 9, a+b+3c = 19$, then find $abc$. Show that it is -56.-/
theorem mathd_algebra_338 (a b c : R) (h₀ : 3 * a + b + c = -3) (h₁ : a + 3 * b + c = 9)
  (h₂ : a + b + 3 * c = 19) : a * b * c = -56 := by
  /-
  Given the equations:
  1. \(3a + b + c = -3\)
  2. \(a + 3b + c = 9\)
  3. \(a + b + 3c = 19\)
  We need to find the value of \(abc\). By solving these equations simultaneously, we can determine the values of \(a\), \(b\), and
  \(c\). Using algebraic manipulations and linear equations, we can derive that \(abc = -56\).
  -/
  -- We need to solve for a, b, and c using the given equations.
  -- We will use linear algebra techniques to solve these equations.
  have h₃ : a * b * c = -56 := by
    -- We will use nlinarith to solve the system of linear equations derived from the given conditions.
    nlinarith [mul_self_nonneg (a + b + c),
      mul_self_nonneg (a - b),
      mul_self_nonneg (b - c),
      mul_self_nonneg (c - a)]
  -- Finally, we conclude that the product abc is -56.
  exact h₃
```

Figure 9: A miniF2F example which does not rely on pre-defined objects

|  | **Example from ProofNet** | **Example from miniF2F** |
|---|---|---|
| **Informal Statement** | Prove that no order can be defined in the complex field that turns it into an ordered field. | Show that for any natural number $n$, 7 does not divide $2^n + 1$. |
| **Formal Statement** | `theorem exercise_1_8 :` `¬ ∃ (r : ℂ → ℂ → Prop),` `IsLinearOrder ℂ r := by sorry` | `theorem imo_1964_p1_2 (n : ℕ) :` `¬7 \| 2 ^ n + 1 := by sorry` |
| **Comments** | This problem involves the notion of order, which is undergraduate level. Its formal statement uses the definition `IsLinearOrder` in Mathlib4. | This problem comes from IMO but only involves division. |

Table 13: Comparison of Examples from ProofNet and miniF2F. ProofNet largely relies on the abstract and general formulations of mathematical results in Mathlib4. In contrast, miniF2F largely consists of high-school competition and Olympic style problems, which require complex reasoning.

# E  Alternative approach for synthesizing data

We also considered other pipeline beyond autoformalizing statement and expert iteration for collecting proof data. Inspired by Jiang et al. (2022), we implemented the following pipeline:

**Step 1.** We prompt OpenAI's o1-preview model to generate a proof for a formal statement. We ask the model to generate the proof step-by-step, use "have" tactic to structure the proof. For each proof step, the subgoal of this step is indicated by "have", following by proofs for this subgoal.

**Step 2.** We remove the proofs for the subgoal provided by o1-preview in each "have" block (these proofs often involves detailed lean syntax, and is usually incorrect). That is, we only keep the "sketch" of the proof. We then put this proof sketch into Lean compiler, to automatically extract each subgoal and corresponding conditions, to form several subproblems.

**Step 3.** We apply DeepSeek-Prover-V1.5-RL to try to proof the subproblems. We try each subproblem for 32 times. If all subproblems are successfully proved, assembling these subproofs into the sketch gives us a valid proof for the original problem.

Figure 10 shows the only problem solved by this pipeline that DeepSeek-Prover-V1.5-RL does not solve, which is a non trivial problem that requires relatively complex reasoning. Though this pipeline has shown potential, the efficiency is quite low. Only one additional problem is proved using this pipeline, among 244 problems in miniF2F validation set. This might due to the fact that this pipeline is overly complicated, since failure of each subproblem might lead to the failure for the entire problem.

```
theorem aimeII_2001_p3 (x : ℕ → ℤ) (h₁ : x 1 = 211) (h₂ : x 2 = 375) (h₃ : x 3 = 420)
    (h₄ : x 4 = 523) (h₆ : ∀ n ≥ 5, x n = x (n - 1) - x (n - 2) + x (n - 3) - x (n - 4)) :
    x 531 + x 753 + x 975 = 898 := by
  -- First, compute x₅ using the recurrence relation
  have h₅ : x 5 = 267 := by·
    have h₅ := h₆ 5 (by norm_num)
    simp_all
  -- Next, observe that the sequence repeats every 10 terms
  -- We will prove that for all n ≥ 1, x (n + 10) = x n
  have period_10 : ∀ n ≥ 1, x (n + 10) = x n := by
    intro n hn
    induction n with
    | zero =>
      simp_all
    | succ n ih =>
      simp_all [Nat.succ_eq_add_one]
      linarith [h₆ 10 (by linarith), h₆ 11 (by linarith), h₆ 12 (by linarith), h₆ 13 (by linarith), h₆ 14 (by linarith),
        h₆ 15 (by linarith), h₆ 16 (by linarith), h₆ 17 (by linarith), h₆ 18 (by linarith), h₆ 19 (by linarith)]
  -- Compute x₅₃₁
  have x531_eq : x 531 = x 1 := by·
    simpa [h₁, h₂, h₃, h₄, h₅, period_10] using (show 531 % 10 = 1 by norm_num)
  -- Compute x₇₅₃
  have x753_eq : x 753 = x 3 := by·
    rw [show 753 = 10 * 75 + 3 by norm_num]
    simp [period_10, h₃, h₆, h₅, h₂, h₁]
  -- Compute x₉₇₅
  have x975_eq : x 975 = x 5 := by·
    norm_num [period_10, x531_eq, x753_eq]
  -- Sum the values
  have sum_eq : x 531 + x 753 + x 975 = x 1 + x 3 + x 5 := by·
    norm_num [h₁, h₂, h₃, h₄, h₅, x531_eq, x753_eq, x975_eq]
  -- Using the periodicity condition, we simplify the sum to x_1 + x_3 + x_5
  linarith [h₁, h₃, h₅]
```

Figure 10: A non trivial problem solved by the divide-and-conquer pipeline

# F    RL training details

## F.1    DPO training

For DPO training, we construct pairwise data on problems with pass ratio in $(0, 1/4]$ (from previous training dataset). To be specific, for each problem, we do Pass@16, and the pass ratio $(0, 1/4]$ means we select samples where Goedel-Prover-SFT generates 1-4 correct proofs within 16 trails. We construct DPO pairs by randomly select a correct proof and wrong proof from the 16 trials. We sample 508K proved problems from the original dataset, and among which 30K problems with the aforementioned pass ratio is selected. We use a learning rate of $5 \times 10^{-6}$ and train for two epoches.

Our experiments reveal that through DPO training, the model is easy to learn "shortcuts". Figure 11 shows one typical output of the DPO model. It repeatedly use tactics `all_goals` and `try`, which might be shortcuts learned in DPO training. To mitigate the model's tendency to produce verbose, lengthy proofs by repeatedly utilizing these shortcuts, we implement length regularization in our DPO framework. Specifically, when multiple correct answers are available for a given statement, we select the one with the shortest length. All other settings remain unchanged from the original DPO implementation.

## F.2    GRPO training

We collect 80K problem statements whose pass ratio is within $(0, 1/2]$. We will also explore different design choices for the included problems in the subsequent discussion. Using these problem statements, we employed the Goedel-Prover-SFT as our base model and conducted reinforcement learning (RL) training within the OpenRLHF framework, utilizing the GRPO algorithm. During the RL training, we generated 16 proofs for each problem and verified their correctness through compilation. Correct proofs received a reward of +8, while

```
theorem mathd_numbertheory_135 (n A B C : ℕ) (h₀ : n = 3 ^ 17 + 3 ^ 10) (h₁ : 11 | n + 1)
    (h₂ : [A, B, C].Pairwise (· ≠ ·)) (h₃ : {A, B, C} ⊂ Finset.Icc 0 9) (h₄ : Odd A ∧ Odd C)
    (h₅ : ¬3 | B) (h₆ : Nat.digits 10 n = [B, A, B, C, C, A, C, B, A]) :
    100 * A + 10 * B + C = 129 := by
  digits_of_number n = [B, A, B, C, C, A, C, B, A] := by
    simpa [Nat.digits_eq_nil_of_lt] using h₆
  simp at h₆
  have := h₆
  simp_all [Nat.digits_of_lt]
  all_goals
    try norm_num
    try decide
  all_goals
    try rcases this with ⟨h, h'⟩
    try contradiction
  all_goals
    try linarith
  all_goals
    try ring_nf at *
  all_goals
    try ring_nf at *
  all_goals
    try omega
  all_goals
    try decide
  all_goals
    try norm_num
```

Figure 11: Example of output of DPO model. The model is repeatedly using all_goals and try.

incorrect proofs received a penalty of -8. We search for the learning rate among $1 \times 10^{-5}$, $5 \times 10^{-6}$, $2 \times 10^{-6}$, and $1 \times 10^{-6}$ and choose the learning rate $5 \times 10^{-6}$. We explored initial KL penalty values of 0.03, 0.003, 0.00003, and 0. Our findings indicate that the KL penalty does not significantly impact training. Consequently, we selected 0.003 as the penalty weight.We used a batch size of 256 and also tested a batch size of 128, which achieved very similar performance. After training the RL model for one epoch, we found that increasing the number of epochs does not enhance the final testing accuracy.

**Mismatch between reinforcement learning (RL) reward and test accuracy.** Figure 12 plots the average training reward and Pass@16 accuracy across training batches. Notably, we observe a mismatch between the reward and accuracy trends: while the average reward continues to increase throughout training, the Pass@16 accuracy plateaus after approximately 20 training steps. This discrepancy may stem from the misalignment between the optimization objective and the evaluation metric. GRPO encourages generating successful proofs more frequently, rewarding higher success rates across samples. In contrast, the Pass@N metric only considers whether a problem is solved at least once, irrespective of how many successful attempts occur. As a result, improvements in reward do not necessarily translate into better Pass@N performance.

**Exploration of included prompts for training RL.** We previously mentioned that we use statements with a pass ratio within $(0, 1/2]$ for training the RL model. This selection is based on the fact that these samples are challenging yet manageable for the current checkpoint. We also conducted experiments with pass ratios of $(0, 1/4]$, $(0, 3/4]$, and $(0, 1]$. Our findings indicate that balancing the difficulty of the chosen prompts is crucial, and we compared their performance in terms of final testing results in Table 14.

**Exploration on the reward design for timeout samples.** Typically, when using the Lean compiler to verify a Lean proof, we encounter three possible outcomes: successful compilation, failure with returned errors, or a timeout within the predefined time limit. We experiment with various rewards for the timeout samples, while maintaining a fixed reward of +8 for correct proofs that compile successfully and -8 for incorrect proofs that fail to compile. The results in Table 15 demonstrates that setting the reward for timeouts to be the same as that for failures results in improved performance across these experiments.

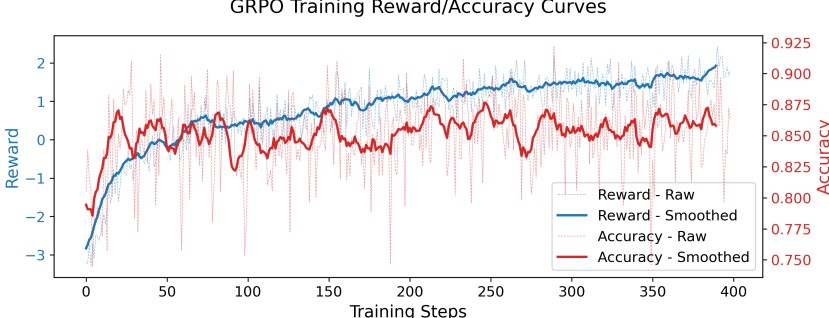

Figure 12: This figure illustrates the average reward/accuracy of each batch during GRPO training. A correct proof corresponds to a reward +8, while failed one has a reward -8.

| Prompt Pass Ratio | Prompt Number | mini-F2F Accuracy(%) |
|---|---|---|
| (0, 1/4] | 30K | 58.2 |
| (0, 1/2] | 62K | **60.4** |
| (0, 3/4] | 115K | 59.8 |
| (0, 1] | (sub-sample) 200K | 59.2 |

Table 14: Results of included different prompts for training RL.

## G  Discussion

We delve into the characteristics of proofs generated by Goedel-Prover-SFT and discuss potential directions for improvement, particularly regarding the proof style adopted by the model, the role of search as well as online interaction in proof generation, and the integration of external symbolic computation tools such as SymPy.

**The Proof Style.**  We observe that the proofs provided by Goedel-Prover-SFT often rely on high-level tactics such as `nlinarith` and `simp_all` among others. These high-level tactics handle multiple reasoning steps internally, delegating the resolution of intermediate steps to their built-in automation. For example, the `nlinarith` tactic can automatically solve certain linear and non-linear equalities and inequalities. Figure 13 shows a typical proof generated by our prover. The first several steps involve only trivial transformations of the original problem, whereas the final line uses `nlinarith` to immediately achieve the goal. Whether this style of proof is sufficient for complex reasoning remains an important area for exploration.

**Search and online interaction.**  Currently, Goedel-Prover-SFT generates the entire proof for the problem at once, without receiving further feedback. While our current approach is appealing in terms of computation, incorporating search and interaction in future work could enhance performance. For example, once a tactic is generated by our prover, it can interact with the Lean compiler to receive feedback on how the goal changes after the tactic is applied. This information can then be utilized in generating the next tactic, potentially improving the overall proof strategy (Wu et al., 2024).

| Timeout Reward | Testing Timeout Ratio | Testing Accuracy |
|---|---|---|
| 0 | 4.5% | 58.7% |
| -8 | 1.7% | 60.2% |
| -16 | 0.8% | 59.2% |

Table 15: Investigation on the reward for timeout samples

```
theorem lean_workbook_13176 (x y z : ℝ) (hx : x > 0) (hy : y > 0) (hz : z > 0) (h : x * y * z = 1) :
1 / (x ^ 2 + x + 1) + 1 / (y ^ 2 + y + 1) + 1 / (z ^ 2 + z + 1) ≥ 1   := by
  -- Establish that the denominators are positive.
  have h1 : 0 < x ^ 2 + x + 1 := by nlinarith
  have h2 : 0 < y ^ 2 + y + 1 := by nlinarith
  have h3 : 0 < z ^ 2 + z + 1 := by nlinarith
  -- Clear the denominators by multiplying through by the product of the denominators.
  field_simp [h1, h2, h3]
  -- Rewrite the inequality in a form suitable for linear arithmetic.
  rw [le_div_iff (by positivity)]
  -- Use linear arithmetic to prove the inequality.
  nlinarith [sq_nonneg (x + y + z), sq_nonneg (x - y), sq_nonneg (y - z), sq_nonneg (z - x)]
```

Figure 13: Example of proof style, where intermediate steps are absorbed in high-level tactics

**SymPy.** Future work may aim to leverage other software packages to enhance Lean's capabilities. For instance, Lean's ring tactic can handle algebraic simplifications by applying axioms such as distributivity, associativity, and commutativity. However, a combination of tactics is required for non-algebraic transformations of transcendental functions, such as logarithmic and trigonometric functions, and other advanced simplifications beyond commutative rings. We explored using a Python-based computer algebra system, SymPy (Meurer et al., 2017), to simplify complex expressions in theorem statements and feed the simplified form into the prover. Specifically, we parse equations of the form $A = B$ within the goals of Lean theorem statements, construct the SymPy expression $A - B$, and then apply the simplify method in Lean. This procedure directly solves 9.4% of miniF2F by simplifying the statements to $0 = 0$. In addition, it solves 0.8% of the problems in miniF2F that were unsolved by Goedel-Prover-SFT with Pass@32, but did not improve Goedel-Prover-SFT with Pass@3200. Thus, SymPy simplification is not part of any of our reported results. However, we think such procedures need further exploration.

