# OpenReview forum: "Goedel-Prover: A Frontier Model for Open-Source Automated Theorem Proving"
_colmweb.org/COLM/2025/Conference — COLM 2025_

### Official Review · Reviewer_Jesv · 2025-04-12

**Rating:** 7
**Confidence:** 5
**Ethics Flag:** 1

**Summary:**

The paper describes Goedel-Prover, a state-of-the-art large language model that can generate a proof for a given Lean 4 theorem. The contribution is two-fold: (1) the pipeline for creating formalization (translating natural language theorems to Lean4) datasets and training the Goedel-Prover model by expert iteration (~rejection sampling), and (2) valuable resources for the field, namely the Goedel-Prover model, a large auto-formalization training data (1.73M), and ~30k proofs for the Lean Workbook theorems (for context, the original Lean Workbook only contains theorems without corresponding proofs).

**Questions To Authors:**

1. **Regarding `try` tactic in RL-trained models** It is quite interesting that models trained with RL use `try` more frequently. Can you observe a similar trial-and-error behavior in the *CoT comments*, e.g., using words like 'wait' or 'alternatively' more often [1]?

2. **Effects of RL in low inference budget.** RL is known to boost performance when sample size (Pass@K) is small, but the gain converges when K gets larger [2]. Indeed, Table 5 shows that the gain by RL is larger in Pass@32 than Pass@3200. How do default and RL-trained models perform in a highly constrained setting, e.g., Pass@1 or Pass@4?

3. **Human eval on Faithfulness & Completeness** I intuitively understand that Claude or other LLMs will do a good job for F&C eval, based on my personal experience. However, I wonder if there is a human evaluation result on a small test set.

[1] Yeo, E., Tong, Y., Niu, M., Neubig, G., & Yue, X. (2025). Demystifying Long Chain-of-Thought Reasoning in LLMs. arXiv preprint arXiv:2502.03373.

[2] Shao, Z., Wang, P., Zhu, Q., Xu, R., Song, J., Bi, X., ... & Guo, D. (2024). Deepseekmath: Pushing the limits of mathematical reasoning in open language models. arXiv preprint arXiv:2402.03300.

**Reasons To Accept:**

1. **Empirical performance.** The performance of Goedel Prover is impressive, with an unprecedented success rate in common benchmarks like MiniF2F and Lean Workbook. The authors have effectively shown that simple strategies like rejection sampling can dramatically boost the performance.
2. **Significant model/data contribution.** Goedel-prover model, ~1.8M proposed formalization data, and ~15k new Lean Workbook proofs will be a fundamental resource for future research in LLM-for-formal-proofs. I personally find Lean Workbook data a significant contribution to the community, since it is currently the largest dataset where one can find natural language questions mapped to their formal proofs.

**Reasons To Reject:**

1. **Technical novelty**. The Goedel-Prover's key methods are (1) translating Numina's abundant NL problems to Lean 4, and (2) applying multiple rounds of rejection sampling to improve the performance. I see the development of Goedel-Prover as a scaled-up version of DeepSeek-Prover v1 (Xin, 2024) with minimal technical innovation. The only novelty of this work I can find is that it uses two formalizer models (Section 3.1) that increase the performance by adding diversity to the input format (Table 3), which is an interesting analysis but significantly underexplored to be a strong contribution. This is the critical reason why I did not give a higher score.

2. **Cannot solve general math problems. (nice-to-have)** Current form of Goedel-Prover takes *formalized theorems* as input, which introduces two limitations.

- (1) It needs autoformalization to prove a theorem given in natural language. Therefore, if the theorem description is given in natural language, the autoformalizer's error can propagate, further reducing the success rate.
- (2) It cannot solve math problems where the answer is not yet known (e.g., *What is the answer?* or *Is this conjecture true or false?*). To solve these problems, the autoformalizer should first find the answer candidate (*The answer is likely 5.*) and prove if that answer is correct (*What is the answer? Prove that it is 5.*).

While these criticisms generally apply to all previous Lean 4 proof generators, it would be nice to explore these new directions with NL-only theorems or open-ended questions. Note: This did not affect the rating a lot.

---

> ### Author Response · Authors · 2025-06-02
> **Reply to Reviewer Jesv**
>
> Thanks for your valuable comments and suggestions. The following is our reply to some of the problems you are concerning:
>
> ## 1. Technical Novelty and Comparison to DeepSeek-Prover v1
>
> > “I see the development of Goedel-Prover as a scaled-up version of DeepSeek-Prover v1 (Xin, 2024) with minimal technical innovation”
>
> We would like to clarify that our work is NOT a scaled-up version of DeepSeek-Prover v1.5. DeepSeek-Prover did not release technical details or any training data, therefore, our pipeline was built **entirely from scratch**. We first constructed a large set of formal statements, and then used these statements to conduct iterative training for the prover. Our primary contribution is not method novelty, but rather the open-source release of our framework and large-scale datasets. This substantial resource contribution lays a solid foundation for the community and pushes the frontier beyond prior closed SOTA works.
>
> ## 2. Limitation: Cannot Solve General Math Problems Directly
>
> We agree with your observation that this is a general limitation of formal proof systems, rather than of our work. Formal proof systems are designed to verify the correctness of proofs for given theorems. Its focus on formal verification enables powerful properties such as proof checking via compilers, which is not possible with natural language. Both formal and natural language approaches have distinct strengths and limitations, and this work aims to advance the formal proof side.
>
> ## 3. Regarding the "try" tactic and trial-and-error behavior in CoT comments
>
> > “It is quite interesting that models trained with RL use try more frequently. Can you observe a similar trial-and-error behavior in the CoT comments, e.g., using words like 'wait' or 'alternatively' more often?”
>
> Thank you for this insightful question. While terms like "wait" or "alternatively" signal self-reflection and advanced reasoning in long chain-of-thought models such as O1 and R1, the situation is quite different in formal provers. In current formal proof models (e.g., Goedel-Prover and Deepseek-Prover-1.5), the CoT comments are typically straightforward and do not exhibit such kind of reflective reasoning. As a result, we can only observe trial-and-error behavior through the usage of formal tactics like try.
>
> The work you cited represents the frontier of long-form natural language reasoning, while our work focuses on formal proof generation. These two areas have not yet been integrated during the time of this work. We appreciate your suggestion and will add a discussion on the potential for combining formal proof generation with advanced chain-of-thought reasoning as a promising direction for future work.
>
> ## 4. RL Effects in Low Inference Budget (Pass@K)
>
> > “Effects of RL in low inference budget. RL is known to boost performance when sample size (Pass@K) is small, but the gain converges when K gets larger [2]. Indeed, Table 5 shows that the gain by RL is larger in Pass@32 than Pass@3200. How do default and RL-trained models perform in a highly constrained setting, e.g., Pass@1 or Pass@4?”
>
> Thank you for this suggestion. We have extended our evaluation to highly constrained settings (Pass@1, Pass@4) for both default and RL-trained models. The results are summarized below:
>
> | Model | Pass@1 | Pass@4 |
> | ----- | ----- | ----- |
> | Default (SFT) | 39.06 | 51.56 |
> | DPO | 37.99 | 52.87 |
> | DPO with length penalty | 41.31 | 53.89 |
> | GRPO | 46.84 | 56.31 |
>
> We will include these results in the revised version.
>
> ## 5. Human Evaluation on Faithfulness & Completeness
>
> > “Human eval on Faithfulness & Completeness I intuitively understand that Claude or other LLMs will do a good job for F&C eval, based on my personal experience. However, I wonder if there is a human evaluation result on a small test set.”
>
> We have conducted a study comparing LLM and human ratings for Faithfulness & Completeness on a subset of 28 examples. The agreement rate is summarized in the table below:
>
> | Human\\Qwen | Correct | False |
> | ----- | ----- | ----- |
> | **Correct** | 12 | 0 |
> | **False** | 4 | 12 |
>
> We will include the detailed evaluation results and discussion in the revised version.

---

> > ### Comment · Reviewer_Jesv · 2025-06-02
> > **Response to authors**
> >
> > Thank you for your detailed response. I generally agree with the authors' response, and most of my questions are fulfilled. Regarding Point 1 (Technical Novelty and Comparison to DeepSeek-Prover v1), I would like to clarify that:
> >
> > - I agree that the open-source dataset contribution (autoformalization and lean-workbook) is substantial in terms of scale, as mentioned in Reasons to publish, which led me to give a high score of 7.
> > - As the authors indirectly agree (by admitting that the method novelty is not the prime claim of the paper), the technical novelty of this work apart from DeepSeek-Prover v1 (note: not v1.5!) is limited.
> >
> > Hence, since there is no major change in my point-of-view, I would like to keep my current evaluation of the paper. Please feel free to initiate further discussion during the author response period if necessary!

---

### Official Review · Reviewer_e6im · 2025-05-07

**Rating:** 6
**Confidence:** 5
**Ethics Flag:** 1

**Summary:**

The paper presents Goedel-Prover, a language model for formal theorem proving in Lean 4. To address the scarcity of formal data, the authors autoformalize 1.64 million informal math problems—primarily from the Numina dataset—using two distinct formalizers, and verify their alignment with a separate LLM. The resulting dataset is used to iteratively fine-tune a base prover model (DeepSeek-Prover-V1.5-Base) through expert iteration. The paper reports consistent performance improvements across iterations, with strong results on benchmarks such as miniF2F, PutnamBench, and the Lean Workbook, particularly in whole-proof generation settings.

**Questions To Authors:**

Most of my questions are covered in the "Reasons to Reject" section. I would be willing to increase my score if the authors can address some of the concerns I have raised.

**Reasons To Accept:**

- The results look quite strong, on miniF2F the model gets 57.6% Pass@32 which is ~7% better than DeepSeek-Prover-v1.5-RL, and also performs best on PutnamBench (at the time of submission, now the record is much higher).
- I liked the way they constructed the dataset — using two different formalizers and also verifying the alignment using another LLM is a solid idea, and probably helps robustness. The scale (1.6M formalized statements) is impressive too.
- The training pipeline is pretty clean. It’s all supervised fine-tuning via expert iteration, no complex RL tricks in the main model. The description is clear and makes sense overall.
- They did some good ablations — like showing the benefit of using diverse formalizations and of scaling the dataset. There’s also some thoughtful analysis on RL overfitting, reward hacking, etc which I appreciated.
- One interesting observation was that applying RL like DPO or GRPO made the model write longer and sometimes repetitive proofs by overusing tactics like try. It’s useful that they highlight these drawbacks instead of just showing the gains — adds some nuance to the field.

**Reasons To Reject:**

- The paper lacks significant novelty. The idea that expert iteration improves performance was already shown in Polu et al. (2022) [arXiv:2202.01344], and it’s well known in general that scaling data helps. This work mainly extends those ideas at scale rather than introducing new techniques or architectural improvements.
- The open-source claim is unverifiable at review time. There is no anonymous link, supplementary code, or dataset. It's unclear whether the entire expert iteration pipeline will actually be released. Prior work often only releases models or inference scripts; without clarity on whether the full training and data generation setup will be included, it's hard to assess how open this effort really is.
- The autoformalization quality checking just uses an LLM to assess alignment. Unlike Jiang et al. (2023) [arXiv:2311.03755], which included human evaluation of a subset of outputs, this work does not conduct any manual or crowd-sourced study. This weakens confidence in the quality of the 1.6M formalized statements.
- The paper does not analyze the quality of the proofs themselves. How many rely on high-level automated tactics like linarith, simp, or ring? How long are the average proofs? Are they interpretable or just exploiting Lean’s built-in automation? These questions are especially relevant given that such tactics can solve many miniF2F problems in one line.
- miniF2F itself is becoming a dated benchmark. Many of the problems can be solved with simple tactics and don't reflect deeper or multi-step mathematical reasoning. It's not clear whether Goedel-Prover performs well because of actual improvements in reasoning or just better exploiting Lean 4’s automation.
- There’s no detailed discussion of potential data leakage. How distinct are the problems in the training dataset from the ones in miniF2F or other benchmarks? Given the overlap with Lean Workbook and other sources, the generalization claims are hard to judge.
- The performance on PutnamBench is only marginally better than prior work (e.g., it solves the same number of problems as ABEL, just with fewer samples), so the improvement is not very substantial outside miniF2F.
- While the observation that RL techniques like DPO and GRPO can lead to longer or degenerate proofs is interesting, it’s more of a side note and not enough to carry the contribution on its own.

---

> ### Author Response · Authors · 2025-06-02
>
> Thank you for your invaluable feedback. We address your suggestions below:
>
> ## 1. Novelty of the Contribution
>
> > “The paper lacks significant novelty. The idea that expert iteration... This work mainly extends those ideas ...”
>
> Our primary contribution is not on methodological novelty, but rather providing an open-source framework and dataset that reproduces and surpasses closed-source SOTA results. We believe this establishes a strong foundation for future research in the field. While expert iteration has been explored before, scaling it successfully to achieve SOTA results with an open framework on a new field represents a significant contribution to the community.
>
> ##  2. Open-source Release and Verifiability
>
> > “The open-source claim is unverifiable at review time.”
>
> I understand your concern. Due to the large size of our complete dataset, we have uploaded 1/4 of the total data to [this anonymous link](https://anonymous.4open.science/r/goedel_rebuttal-4ACB). It includes both formal statements and proofs.
>
> ## 3. Quality of Autoformalization: Human Evaluation
>
> > “The autoformalization quality checking just uses an LLM to assess alignment...”
>
> Yes! We did conduct human evaluations on a randomly sampled subset of our autoformalized statements. The results show 85.7% agreement, as summarized below:
>
> | Human\\Qwen | Correct | False |
> | ----- | ----- | ----- |
> | **Correct** | 12 | 0 |
> | **False** | 4 | 12 |
>
> We will add these results and further explanation to the revised paper.
>
> ## 4. Analysis of Proof Quality and Use of High-level Tactics
>
> > “...How many rely on high-level automated tactics like linarith, simp, or ring? How long are the average proofs?”
>
> We appreciate this feedback. In the paper, we already analyze proof length and the use of "try" tactics across different models in Table 5 in the main text. We add additional statistics on the usage of high-level tactics like linarith, simp, and ring.
>
> | Model | linarith | simp | ring | length |
> | ----- | ----- | ----- | ----- | ----- |
> | Goedel-Prover-SFT | 1.25 | 2.12 | 0.53 | 369 |
> | Goedel-Prover-DPO | 2.57 | 3.24 | 0.96 | 450 |
> | Goedel-Prover-RL | 1.50 | 2.57 | 1.42 | 426 |
> | DeepSeek-Prover-1.5-RL | 1.24 | 2.15 | 0.49 | 364 |
>
> > “Are they interpretable or just exploiting Lean’s built-in automation?”
>
> We would also like to emphasize that high-level Lean tactics are not trivial and requires significant reasoning. The model must decide where and how to apply these tactics, e.g., which theorems to use for simplification. This process is interpretable and demonstrates rigorous reasoning, rather than merely exploiting Lean’s automation.
>
> > “... such tactics can solve many miniF2F problems in one line.”
>
> Regarding miniF2F, while it contains some solvable problems via a single tactic, it also includes highly challenging questions (e.g., IMO problems). Our improvements over baselines are particularly notable on these more difficult problems, not just the simpler ones.
>
> ## 5. Benchmark Relevance and Breadth, Comparative Performance on PutnamBench
>
> > “miniF2F itself is becoming a dated benchmark....”
>
> > “The performance on PutnamBench... the improvement is not very substantial outside miniF2F.”
>
> We respectfully disagree that miniF2F is a dated benchmark. It remains a widely-used and challenging dataset in the field. Although some of the problems in miniF2F can be solved with simple tactics, it has a steep difficulty curve and a substantial number of multi-step, high-complexity problems, e.g., AIME/IMO problems. Meanwhile, as shown in Lines 203-227, our model achieves consistent improvements not only on miniF2F, but also on PutnamBench and NuminaTest, demonstrating broad capabilities across different benchmarks.
>
> Furthermore, while ABEL approaches our model’s performance on PutnamBench, it achieves only 41.3% on miniF2F, which is much lower than ours. Taken across all datasets, our model demonstrates the strongest overall capability.
>
> > “It's not clear whether Goedel-Prover performs well because of actual improvements in reasoning or just better exploiting Lean 4’s automation.”
>
> The distinction between "reasoning improvement" and "exploiting Lean 4's automation" creates a false dichotomy in formal proving. Effectively leveraging Lean's automation is itself a sophisticated reasoning capability requiring deep understanding of both the mathematical problem and the proof assistant. Improved reasoning and effective automation use are complementary strengths that together produce a better prover.
>
> ## 6. Data Leakage and Overlap with Benchmarks
>
> > “There’s no detailed discussion of potential data leakage....”
>
> We have already taken careful steps to prevent data leakage. Specifically, we extracted and compared statement content and removed any exact matches between our training set and the evaluation benchmarks. We thank you for raising this point and will clarify our methodology in the revision to ensure transparency regarding data splits and overlap prevention.

---

> > ### Comment · Reviewer_e6im · 2025-06-05
> >
> > Thank you for your detailed response. I greatly appreciate your efforts to clarify my doubts.
> >
> > 1. After reading your response it seems that one major contribution is the dataset itself (rather than novelty of the approach itself), and hence it becomes more important to share some insights about the dataset itself. I appreciate that you shared anonymous links, it helped me see some of the formalizations.
> >
> > 2. I appriciate the authors doing some quality assesment, however, 28 data-points is too small to be statistically significant. For reference please see Jiang et al. (2023) [arXiv:2311.03755], they do random sampling and case studies for about 600 data points (their original dataset was 100k data poitns), 28 is very low compared to that.
> >
> > 3. It is really important to know the reliance on automated tactics like `linarith`, `nlinarith`, `simp`, etc. These tactics get much better with new versions of Lean and hence things become uncomparable between different approaches that are evaluating on different Lean versions. It is also not hard to for models to actually heavily rely on these automations and learn to predict them in most the proofs sampled and thus increasing the likelihood to success without really learning deeper semantics.
> >
> > 4. I agree that miniF2F has some IMO problems (which can be considered hard), however, how many IMO problems did Goedel Prover solve? Any reason you didn't choose to highlight those IMO problem proofs in your paper (assuming your approach solved those hard miniF2F problems). I also believe miniF2F has some really hard induction problems, did you solve any of those? As mentioned earlier, a lot heavy lifting can be done via the automated tactics, for example in Fig. 11, a lot heavy lifting is actually done by `norm_num`.
> >
> > 5. Exact match is a weak measure for data leakage (given the proofs can be very similar, and problem statements can be very similar like LHS swapped with RHS). I would have loved to see more case studies and more experiments around trying to look up similar problems in train set which might have inspired some test solutions.
> >
> > In general, I find it a big weakness in a dataset paper with limited assesement about the data quality, train, test splits and potential data leakage issues.
> >
> > Leaving all that aside, I belive that authors do make some nice contribution in terms of open sourcing their model and dataset. It takes a lot of engineering effort to build something like this, so I will increase my score. It is very tough call between a weak accepting or weak rejecting. However, I give benefit of doubt to the authors for their effort. However, I must mention the work, experiments, and case studies are far from complete.

---

> > ### Author Response · Authors · 2025-06-11
> > **Follow-up Response （1/2）**
> >
> > ### 1. Contribution and Dataset
> >
> > We appreciate your recognition that our open-source framework and dataset are major contributions. While the dataset itself is a core deliverable, we want to emphasize that the process of constructing this dataset—especially the formalization of statements and iterative training of provers—is nontrivial, particularly when starting from scratch in a new domain. Both the autoformalization pipeline and the iterative expert training require significant engineering and design choices, which we describe in detail in the paper. Following your suggestion, we will add a dedicated section in the revised version to analyze the features and unique aspects of our dataset in depth.
> >
> > ### 2. Human Evaluation of Autoformalization
> > While Jiang et al. (2023) performed human evaluation directly on their contributed dataset, our approach was to first assess the reliability of the Qwen on formalization judgments through human evaluation (achieving 85.7% agreement), and then leverage Qwen for large-scale annotation. This two-stage process is different, and we found that these assessments were robust enough to justify their use for large-scale labeling. We agree that expanding the human evaluation set would further strengthen our results and will consider this for future work.
> >
> > ### 3. Problem Difficulty of Minif2f
> > To reflect problem difficulty, we analyzed proof lengths for both our model and the baseline:
> >
> > | Model                  | Total | 1 line | 2-3 lines | >3 lines | amc Problems | aime Problems |
> > |------------------------|-------|--------|-----------|----------|--------------|---------------|
> > | DeepSeek-Prover-V1.5-RL | 123   | 6      | 31        | 86       | 11           | 4             |
> > | Goedel-Prover-SFT       | 142   | 10     | 37        | 95       | 14           | 5             |
> >
> > Goedel-Prover-SFT not only solves more problems, but also produces more multi-step proofs and solves more challenging AMC and AIME problems. This indicates that our model is tackling more complex problems that require deeper reasoning, rather than relying on single-line, automated tactics.
> >
> > ### 4. Usage of Diverse Tactics
> > We also examined the diversity and complexity of tactics used:
> >
> > | Model                  | Total Proofs | Total Tactics | Unique Tactics |
> > |------------------------|-------------:|--------------:|---------------:|
> > | DeepSeek-Prover-V1.5-RL | 123          | 1019          | 112            |
> > | Goedel-Prover-SFT       | 142          | 1190          | 158            |
> >
> > Goedel-Prover-SFT uses a broader and more diverse set of tactics, demonstrating an ability to address a wider range of mathematical challenges rather than overfitting to simpler, formulaic solutions.
> > Overall, our analysis demonstrates that Goedel-Prover-SFT is not merely benefiting from automation, but is solving more multi-step and challenging problems, using a wider array of tactics and more interpretable proof construction. We believe this addresses concerns about both the depth of reasoning and the potential overuse of automation.

---

> ### Comment · Area_Chair_N6G4 · 2025-06-05
>
> Dear reviewer,
>
> Do you have any responses to the authors' rebuttals? Since there is a divergence of ratings among reviewers, it would be very helpful for us to hear your thoughts on whether the rebuttals sufficiently address your concerns.
>
> -AC

---

> > ### Comment · Reviewer_e6im · 2025-06-05
> >
> > I gave my detailed response to the author's rebuttal. I still believe that this paper can be made much better, I have described my reasons in my latest comments.

---

> ### Author Response · Authors · 2025-06-11
> **Follow-up Response (2/2)**
>
> ### 5. Top Tactics Used
> To address concerns about over-reliance on high-level automation, we provide below the top 10 most-used tactics:
>
> | Rank | DeepSeek-Prover-V1.5-RL | Count | %     | Goedel-Prover-SFT | Count | %     |
> |------|--------------------------|-------|-------|--------------------|-------|-------|
> | 1    | `norm_num`               | 193   | 18.9% | `have`             | 200   | 16.8% |
> | 2    | `have`                   | 121   | 11.9% | `linarith`         | 116   | 9.7%  |
> | 3    | `linarith`               | 94    | 9.2%  | `norm_num`         | 96    | 8.1%  |
> | 4    | `simp`                   | 52    | 5.1%  | `simp_all`         | 54    | 4.5%  |
> | 5    | `decide`                 | 42    | 4.1%  | `simp`             | 49    | 4.1%  |
> | 6    | `simp_all`               | 35    | 3.4%  | `only`             | 44    | 3.7%  |
> | 7    | `omega`                  | 30    | 2.9%  | `omega`            | 36    | 3.0%  |
> | 8    | `only`                   | 30    | 2.9%  | `nlinarith`        | 32    | 2.7%  |
> | 9    | `with`                   | 25    | 2.5%  | `ring_nf`          | 28    | 2.4%  |
> | 10   | `nlinarith`              | 23    | 2.3%  | `Finset`           | 27    | 2.3%  |
>
> Unlike the baseline, which heavily favors high-level automation tactics like norm_num, our model’s most frequently used tactic is have, which depends on generating intermediate lemmas and thus requires active construction and decomposition of proofs. This indicates that Goedel-Prover-SFT is not simply overusing automated tactics, but is capable of producing more interpretable and structured proofs.
>
> ### 6. Frequency of Common Automation Tactics
> We also directly measured the use of high-level automation tactics across models:
>
> | Tactic    | DeepSeek-Prover-V1.5-RL |                   |                | Goedel-Prover-SFT |                   |                |
> |-----------|------------------------:|------------------:|----------------:|------------------:|------------------:|----------------:|
> |           | Count                   | % of Proofs       | Avg per Proof   | Count             | % of Proofs       | Avg per Proof   |
> | `aesop`   | 0                       | 0%                | 0               | 1                 | 0.7%              | 0.01            |
> | `decide`  | 42                      | 8.1%              | 0.34            | 21                | 4.9%              | 0.15            |
> | `linarith`| 94                      | 40.7%             | 0.76            | 116               | 33.1%             | 0.82            |
> | `nlinarith`| 23                     | 13.0%             | 0.19            | 32                | 19.0%             | 0.23            |
> | `norm_num`| 193                     | 46.3%             | 1.57            | 96                | 43.7%             | 0.68            |
> | `omega`   | 30                      | 17.1%             | 0.24            | 36                | 20.4%             | 0.25            |
> | `ring`    | 11                      | 8.1%              | 0.09            | 20                | 9.9%              | 0.14            |
> | `simp`    | 52                      | 30.1%             | 0.42            | 49                | 26.1%             | 0.35            |
>
>
> For most high-level tactics, Goedel-Prover-SFT either uses them less frequently or at a similar rate as the baseline. This suggests that Goedel-Prover-SFT does not achieve its improved performance simply by relying on automation, but through more nuanced and varied proof strategies.

---

### Official Review · Reviewer_r5wC · 2025-05-12

**Rating:** 7
**Confidence:** 4
**Ethics Flag:** 1

**Summary:**

The paper tackles the problem of automated theorem proving by introducing a dataset of formal statements and proofs built by iteratively training a series of models. The authors introduce a model trained on this dataset, a finetuned version of deepseek-prover-1.5-base. The authors compare this model to RL-tuned and SFT versions of the base model.

**Questions To Authors:**

- What happens when you switch the Formalizer B model for an open source model instead of claude?
- Can you benchmark training with a different base model?
- Can you also compare size of the models or flops in table 2?

**Reasons To Accept:**

- The paper is clearly written and well motivated
- The list of benchmarks tested is pretty comprehensive
- Across different benchmarks, the Goedel prover outperforms other models
- The experiments with the training recipe were particularly good:

    1) I liked the analysis on scaling the size of the statement set.

    2) correlation between the datasets was also pretty insightful.

    3) The analysis RL training (GRPO, DPO) and alternate proof generation strategies do a good job of making the paper more complete.

**Reasons To Reject:**

- A huge part of the synthetic data generation pipeline relies on closed models.
- Only one base model is used for most experiments in the paper.
- Rater reliability for the FC test is missing. It would be good to see how much Qwen and humans agree.
- I know that the space is a little limited, but it would be useful to move the discussion into the main section of the paper.

**Minor:**

- Missing citations for expert iteration:
    - STaR Bootstrapping reasoning with reasoning
    - Reinforced Self-Training (ReST) for Language Modeling
    - Training Chain-of-Thought via Latent-Variable Inference

---

> ### Author Response · Authors · 2025-06-02
> **Reply to Reviewer r5wC**
>
> Thanks for your valuable comments and suggestions. The following is our reply to some of the problems you are concerning.
>
> ## 1. Reliance on Closed Models in Synthetic Data Generation
>
> > “A huge part of the synthetic data generation pipeline relies on closed models.”
>
> We only used closed models for the minimal initial bootstrapping of the Formalizer, not for large-scale data generation. Specifically, we used Claude to annotate 170k statement translations, trained two Formalizers with this data, and then used these Formalizers to annotate 2M examples. Meanwhile, the Prover training was solely based on open-source models. Thus, only the minimal necessary annotation step relied on closed models, and the scaling was achieved with open-source models. Additionally, the closed model annotation costs were minimal compared to the overall GPU training expenses.
> Furthermore, all formalized statements and proofs, which are generated entirely with open-source models, are now open-sourced as valuable community resources.
>
> ## 2. Only One Base Model Used for Experiments / Benchmarking Other Base Models
>
> > “Only one base model is used for most experiments in the paper.” “Can you benchmark training with a different base model?”
>
> We chose Deepseek-Prover-1.5-Base as our base model to ensure a fair and direct comparison with the most relevant baseline and previous SOTA, Deepseek-Prover-1.5, which also only used this base. Since our improvements primarily stem from better supervised data rather than algorithmic changes, we expect limited sensitivity to the base model. Thus, benchmarking with other base models is not essential to our main claims.
>
> ## 3. Rater Reliability for the FC Test
>
> > “Rater reliability for the FC test is missing. It would be good to see how much Qwen and humans agree.”
>
> Yes! We have sampled 28 FC test problems and compared Qwen with human ratings, finding an agreement rate of 85.7%. We will include these results in the revised version.
>
> | Human\\Qwen | Correct | False |
> | ----- | ----- | ----- |
> | **Correct** | 12 | 0 |
> | **False** | 4 | 12 |
>
> ## 4. Moving Discussion into the Main Section
>
> > “I know that the space is a little limited, but it would be useful to move the discussion into the main section of the paper.”
>
> We appreciate your suggestion and will summarize and incorporate relevant discussion from the appendix into the main sections in the revised paper.
>
> ## 5. Switching to Formalizer B
>
> > “What happens when you switch the Formalizer B model for an open source model instead of claude?”
>
> As noted above, annotation based on closed-source models was only used for the initial seed data for Formalizer. After training on Claude-annotated data, our Formalizer was able to scale up translation for the prover training data. We will clarify it in our revision.
>
> ## 6. Model Size Comparison in Table 2
>
> > “Can you also compare size of the models or flops in table 2?”
>
> Thank you for the suggestion. We add a model size comparison of models in Table 2 as requested.
>
> | Model | Size  |
> | ----- | ----- |
> | Goedel-Prover-SFT | 7B |
> | ABEL (Gloeckle et al.) | 8B |
> | InternLM2.5-StepProver (Wu et al., 2024\) | 7B |
> | InternLM 7B (Ying et al., 2024b) | 7B |
> | GPT-4o | N/A (API) |
> | COPRA (GPT-4o) (Thakur et al., 2023\) | N/A (API) |
> | ReProver w/ retrieval (Yang et al., 2024b) | 0.6B |
> | ReProver w/o retrieval (Yang et al., 2024b) | 0.6B |
>
> We will add the column to Table 2 in our revision.
>
> ## 7. Missing Citations for Expert Iteration
>
> Thank you for the suggestions and we will add these references in the related work section in our revision.

---

> > ### Comment · Reviewer_r5wC · 2025-06-03
> >
> > Thank you for your response!
> > Most of my main concerns related to the use of closed-source models and experiments with just one base model aren't satisfied. I will keep my current score.

---

### Official Review · Reviewer_Zigm · 2025-05-13

**Rating:** 8
**Confidence:** 2
**Ethics Flag:** 1

**Summary:**

This paper describes a methodology for iteratively augmenting a corpus of informal (= natural language) mathematical proofs, using a formalizer (trained on informal/formal proof pairs), a prover, and a verifier together. This augmentation process allows their prover to achieve new state of the art results on several mathematical reasoning benchmarks.

**Reasons To Accept:**

This is an impressive engineering effort that leads to impressive results.

**Reasons To Reject:**

I don't have major concerns (this is outside my area of expertise).

---

> ### Author Response · Authors · 2025-06-02
> **Reply to Reviewer Zigm**
>
> We appreciate your recognition of our methodology and contributions. If you have any further questions during the rest of the rebuttal period, we would be happy to discuss them with you.

---

### Decision · Program_Chairs · 2025-07-08

**Decision:**

Accept

**Comment:**

Thanks to the authors/reviewers for their engagement during the discussion period!

_Quality_: The approach proposed in the work yields solid empricial results (all reviewers), is comprehensively evaluated across a range of benchmarks (Reviewer r5wC), and is accompanied with interesting analyses and ablations (Reviewers r5wC and e6im). Reviewers r5wC, e6im, and Jesv pointed out the need for (small-scale) human evaluation, which the authors provided during the discussion. The amount of datapoints analyzed was small, which is understandable given the time constraints, but I agree with Reviewer e6im that this eventually needs to be scaled up at least slightly beyond n=28. The authors have also followed up with several additional analyses following reviewer requests, and these will be nice additions to include in the subsequent revision.

_Clarity_: No major clarity issues were raised and several reviewer questions were address adequately during the discussion period. Further clarity on which parts of the pipeline relies on closed source models and how the proposed approach relates to prior work would be helpful to have in the subsequent draft (most of this has already been discussed here).

_Originality_: While a few reviewers questioned the methodological originality of the work, I agree with the authors that an open source release of the training/data recipe and the dataset itself, combined with strong empirical results (as acknowledged by all reviewers), is a valuable original contribution.

_Overall significance_: Solid paper with on a topic of interest to the COLM audience with useful open source contributions.

[Automatically added comment] At least one review was discounted during the decision process due to quality]